SOFTWARE

# iModulonMiner and PyModulon: Software for unsupervised mining of gene expression compendia

Anand V. Sastry[1‡], Yuan Yuan[1‡], Saugat Poudel[1], Kevin Rychel[1], Reo Yoo[1], Cameron R. Lamoureux[1], Gaoyuan Li[1], Joshua T. Burrows[1], Siddharth Chauhan[1], Zachary B. Haiman[1], Tahani Al Bulushi[1], Yara Seif[1], Bernhard O. Palsson[1,2,3,4], Daniel C. Zielinski[1] *

1 Department of Bioengineering, University of California, San Diego, La Jolla, California, United States of America, 2 Bioinformatics and Systems Biology Program, University of California, San Diego, La Jolla, California, United States of America, 3 Department of Pediatrics, University of California, San Diego, La Jolla, California, United States of America, 4 Novo Nordisk Foundation Center for Biosustainability, Technical University of Denmark, Kemitorvet, Kongens, Lyngby, Denmark

‡ These authors share first authorship on this work.
* dczielin@ucsd.edu

**Data Availability Statement:** Code and requisite data is made available on Github at https://github.com/sbrg/iModulonMiner/ and https://github.com/SBRG/pymodulon.

## Abstract

Public gene expression databases are a rapidly expanding resource of organism responses to diverse perturbations, presenting both an opportunity and a challenge for bioinformatics workflows to extract actionable knowledge of transcription regulatory network function. Here, we introduce a five-step computational pipeline, called iModulonMiner, to compile, process, curate, analyze, and characterize the totality of RNA-seq data for a given organism or cell type. This workflow is centered around the data-driven computation of co-regulated gene sets using Independent Component Analysis, called iModulons, which have been shown to have broad applications. As a demonstration, we applied this workflow to generate the iModulon structure of *Bacillus subtilis* using all high-quality, publicly-available RNA-seq data. Using this structure, we predicted regulatory interactions for multiple transcription factors, identified groups of co-expressed genes that are putatively regulated by undiscovered transcription factors, and predicted properties of a recently discovered single-subunit phage RNA polymerase. We also present a Python package, PyModulon, with functions to characterize, visualize, and explore computed iModulons. The pipeline, available at https://github.com/SBRG/iModulonMiner, can be readily applied to diverse organisms to gain a rapid understanding of their transcriptional regulatory network structure and condition-specific activity.

## Introduction

Over the past few decades, advances in sequencing technologies have led to a rapid increase in the availability of public transcriptomic datasets [1,2]. Integrative analyses of these public expression datasets has resulted in a comprehensive view of organism transcriptomic states

**Funding:** This work was funded by the Novo Nordisk Foundation Center for Biosustainability and the Technical University of Denmark (grant number NNF20CC0035580 to BOP). The funders had no role in study design, data collection and analysis, decision to publish, or preparation of the manuscript.

**Competing interests:** The authors declare that they have no competing interests.

[3,4], the generation of new biological hypotheses [5,6], and inference of co-expression networks and transcriptional regulation [7,8].

Independent Component Analysis (ICA) has proven to be a powerful method to extract knowledge from large transcriptomics compendia [9–16]. ICA is a machine learning algorithm designed to separate mixed signals into their original source components, based on the equation $X = MA$, where $X$ is the data matrix, $M$ is the components matrix (sometimes called $S$ in the literature for 'sources'), and $A$ is the activities matrix [17].

In the context of the transcriptome, ICA can be applied to transcriptomics datasets to extract gene modules whose gene membership is statistically independent to other modules. The components $M$ calculated by ICA are independently modulated groups of genes, and thus have been termed *iModulons*. Many iModulons are consistent with regulons, or groups of genes regulated by the same transcriptional regulator, in model bacteria [9,10]. iModulons can be genetically observed through binding sites at gene promoters in many cases [18], and can be used to discover new regulons or gene functions in less-characterized organisms [11,19]. The activities matrix $A$ contains the condition-specific activation levels of each iModulon. For regulator-associated iModulons, they represent the activity states of the corresponding transcriptional regulator. iModulon activities have intuitive interpretations, and together with the components $M$ comprises a data-drive approximation of the structure and activity of an organism's transcriptional regulatory network (TRN) [20–22].

iModulons have many properties that lend themselves to knowledge generation from large datasets. ICA outcompeted 42 other regulatory module detection algorithms, including WGCNA and biclustering algorithms, in detecting known regulons across *E. coli*, yeast, and human transcriptomics data [23]. Independent components have been shown to be conserved across different datasets [24,25], batches [26] and dimensionalities within the same dataset [27,28]. ICA has been applied now to a large number of microbial organisms [10,11,21,29–37], demonstrating iModulon analysis as a powerful tool to interpret the ocean of publicly available transcriptomic data to advance our understanding of transcriptome organization.

We have outlined a five-step workflow, called iModulonMiner (https://github.com/SBRG/iModulonMiner), that enables researchers to build and characterize the iModulon structure for any organism or cell type with sufficient public data (**Fig 1A**). The first two steps are to download and process all publicly available RNA-seq data for a given organism. Third, the data must be inspected to ensure quality, and curated to include all appropriate metadata. Next, ICA can be applied to the high-quality compendium to produce independent components. Finally, the independent components are processed into iModulons and can subsequently be characterized. To facilitate iModulon characterization, interpretation, and visualization, we also present PyModulon, a Python library for downstream iModulon analysis (https://pymodulon.readthedocs.io/en/latest/).

## Design and implementation

### Step 1: Compiling all public transcriptomics data for an organism

The NCBI Sequence Read Archive (SRA) is a public repository for sequencing data that is partnered with the EMBL European Nucleotide Archive (ENA) and the DNA Databank of Japan (DDBJ) [38]. We provide a script (1_download_metadata/download_metadata.sh) that uses Entrez Direct [39] to search for all public RNA-seq datasets on SRA and compile annotated metadata into a single tab-separated file. Missing metadata is manually extracted from corresponding literature (**Supplementary Methods in S1 Text**). Each row in the file corresponds to a single experiment, and users may manually add private datasets.

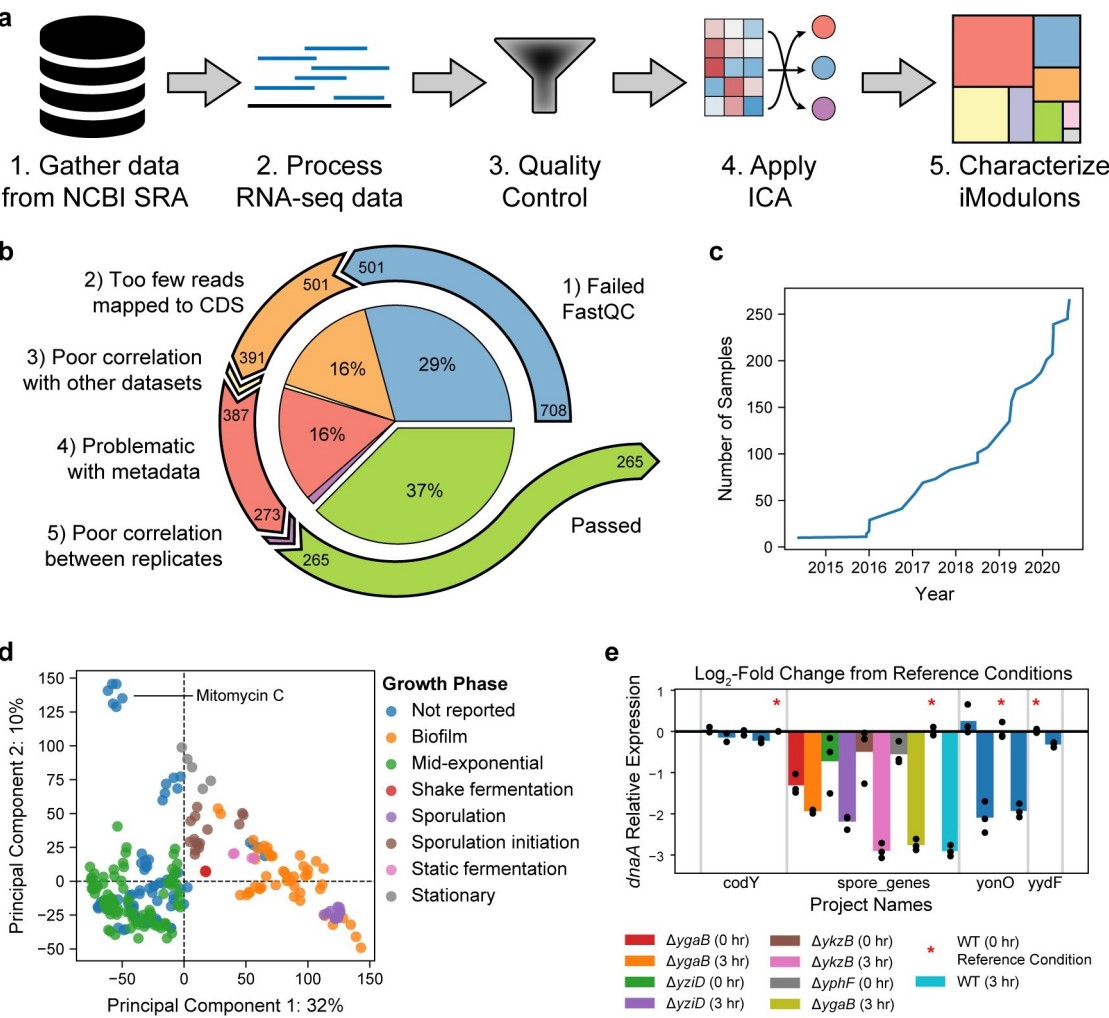

**Fig 1. Introduction to the iModulonMiner using public *B. subtilis* RNA-seq data as a case study. a)** Graphical representation of the five step workflow. **b)** Pie chart illustrating the quality control process. Numbers at the beginning of arrows represent the number of datasets before the quality control step, and numbers at the end represent the number of passed datasets after the step. **c)** Number of high-quality RNA-seq datasets for *B. subtilis* in NCBI SRA over time. **d)** Scatter plot of the top two principal components of the *B. subtilis* expression compendium. Points are colored based on the growth phase parsed from the literature. **e)** Bar chart showing the expression of *dnaA* across four projects. Points show individual replicates, while bars show the average expression for a given condition. Bars with a red star serve as the reference condition for the project.

Although iModulons can be computed from other expression data types, including microarray, RNA-seq, and proteomics, microarray datasets tend to produce more uncharacterized iModulons and induce stronger batch effects through platform heterogeneity [24], and proteomics typically has reduced coverage. For these reasons, we have designed the first two steps specifically for compiling and processing RNA-seq data. We recommend having RNA-seq data for at least 50 unique conditions for an organism before proceeding with the remaining pipeline.

## Step 2: Processing RNA-seq data

Users have the flexibility to select their preferred RNA-seq processing pipeline. Alternatively, they can follow the pipelines listed in https://github.com/SBRG/iModulonMiner/tree/main/2_process_data. For prokaryotic data, the tab-separated metadata file can be directly piped into

the prokaryotic RNA-seq processing pipeline implemented using Nextflow v22.10.0 [40] for reproducibility and scalability. The first step in the pipeline is to download the raw FASTQ files from NCBI using fasterq-dump (https://github.com/ncbi/sra-tools/wiki/HowTo:-fasterq-dump). Next, read trimming is performed using Trim Galore (https://www.bioinformatics.babraham.ac.uk/projects/trim_galore/) with the default options, followed by FastQC (http://www.bioinformatics.babraham.ac.uk/projects/fastqc/) on the trimmed reads. Next, reads are aligned to the genome using Bowtie [41]. The read direction is inferred using RSEQC [42] before generating read counts using featureCounts [43]. Finally, all quality control metrics are compiled using MultiQC [44] and the final expression dataset is reported in units of log-transformed Transcripts per Million (log-TPM). Raw counts are also saved and the median of ratios could be used for following steps as well. In our experience for *E. coli*, the use of normalized counts and TPM yield very similar results. It is expected that in eukaryotes, the differences will be more significant (**Note A in S1 Text**). The Nextflow pipeline can be run locally or on high-performance computing such as Amazon Web Services (AWS).

### Step 3: Quality control and data normalization

To guarantee a high quality expression dataset, data that failed any of the following FASTQC metrics are discarded: per base sequence quality, per sequence quality scores, per base n content, and adapter content. Samples that contain under 500,000 reads mapped to coding sequences are also discarded. Hierarchical clustering is used to identify samples that do not conform to a typical expression profile, which is a criteria for exclusion [3].

Manual curation of metadata is also performed to include experimental details of the samples. These include the strain of the sample, culture media, temperature, growth phase, and relevant experimental information that can facilitate downstream iModulon characterization. The recommended curations can be found in the step3 instructions (https://github.com/SBRG/iModulonMiner/blob/main/3_quality_control/expression_QC_part1.ipynb).

To obviate any batch effects resulting from combining different expression datasets, reference conditions are selected within each project to normalize each dataset. This ensures that nearly all independent components are due to biological variation, rather than technical variation. When choosing this type of normalization as opposed to a single global reference condition, gene expression and iModulon activities can only be compared within a project to a reference condition, rather than across projects.

### Step 4: Computing the optimal number of robust independent components

To compute the optimal independent components, an extension of ICA is performed on the RNA-seq dataset as described in McConn et al. [48].

Briefly, FastICA [49] (scikit-learn v1.0.2 [50]) is executed repeatedly with varying random seeds. A timeout mechanism is implemented to prevent indefinite waiting from potential convergence issues within the algorithm. The resulting independent components are clustered using DBSCAN [51] (scikit-learn v1.0.2 [50]) to identify robust components, using a maximum distance threshold of 0.1 and a minimum cluster size of half the number of FastICA executions. To account for identical components with opposite signs, the following distance metric is used for computing the distance matrix:

$$d_{x,y} = 1 - ||\rho_{x,y}||$$

where $\rho_{x,y}$ is the Pearson correlation between components $x$ and $y$. The final robust components are defined as the centroids of the clusters.

Since the number of dimensions in ICA can alter the results, we apply the above procedure to the compendia multiple times, ranging the number of dimensions from 10 to the nearest multiple of 10 below the sample count, in steps of 10. To identify the optimal dimensionality, we compare the number of independent components (ICs) with single genes to the number of ICs that are correlated (Pearson correlation > 0.7) with the ICs in the largest dimension (called "final components"). We select the number of dimensions where the number of non-single gene ICs is equal to the number of final components in that dimension (**Fig B in S1 Text**).

The iModulons from different subsets of the dataset converge as the number of unique conditions increases (**Fig C in S1 Text**). Once an iModulon structure has been defined for an organism, this structure can be inverted to infer iModulon activities for new transcriptional dataset without re-running ICA, using the function *infer_activities* (**Supplementary Results in S1 Text**).

## Step 5: Characterizing, annotating and visualizing iModulon results with PyModulon

To facilitate the analysis and understanding of iModulons, we have developed the PyModulon Python package to streamline the process of downstream iModulon analysis (https://github.com/SBRG/pymodulon). PyModulon offers a suite of tools that enable researchers to explore, visualize, and gain insights into the complex relationships and patterns within iModulons.

At the core of the PyModulon package is the *IcaData* object. The object contains all data related to iModulons for a given dataset, including the **M** and **A** matrices (**Note B in S1 Text**), the expression matrix, a draft TRN mined from literature, and thresholds used to define iModulons gene membership. Through PyModulon, users can delve into various aspects of the *IcaData* object for iModulon mining. This includes exploring iModulons through gene annotations and functional enrichments, visualizing the iModulons and their activities using a variety of plotting functions, performing motif search, clustering iModulon activities, and creating interactive dashboards for the organism of interest on iModulonDB.org. Furthermore, PyModulon offers functionalities that allow researchers to compare iModulon structures across organisms and estimate iModulon activities for external datasets. The comprehensive list of PyModulon's functionalities can be found at https://pymodulon.readthedocs.io/en/latest/. For more detailed information on the implementation of these functionalities, please refer to the **Supplementary Methods in S1 Text**.

## Workflow alternatives

Several steps of the workflow can be replaced by alternative methods. For example, an alternative processing workflow for eukaryotic RNA-seq data is available in nf-core (https://nf-co.re/rnaseq) [45]. Public or private data can be collected and aligned using the nf-core workflow. Alignment and quantification options include tools such as STAR [46], however pseudo alignment using Salmon [47] is viable for the generation of count matrices and TPM values, as they are necessary for running ICA while BAM alignment files are not. Standard parameters for nf-core alignment for read trimming and feature counts can be used. Suggested alternatives for processing and quality control of the data have been included in the workflow documentation. Most functions for analysis of data in PyModulon are effective regardless of organism type; however several functions specific to data processing and analysis of eukaryotic organisms have been added to PyModulon, and the usage of these functions are demonstrated in the iModulonMiner workflow. For an example of analysis and results for a *S. cerevisiae* dataset, please

refer to the **Supplementary Results in S1 Text**. Additionally, possible alternatives to the fastICA algorithm used in Step 4 are suggested in the README file on the GitHub repository.

## Workflow computational performance

**Data processing times.** Downloading experimental metadata using esearch and efetch typically takes only a few minutes ($<$ 10 minutes for 15,000 *E. coli* samples). fasterq-dump with Nextflow is used to download and stage the RNA-Seq data in parallel with other tasks. The Nextflow RNA-Seq processing pipeline typically takes a few hours for several hundred samples.

**ICA computational requirements.** We evaluated the computational requirements for this analysis on the *E. coli* PRECISE-1K RNA-Seq compendium consisting of 4257 genes and 1035 samples, with an input gene expression log2(TPM) matrix of 80.2 MB. The machine used had the following specifications: CPU: AMD Ryzen Threadripper PRO 5995WX (256 MB cache, 64 cores, 128 threads), RAM: 256GB, DDR4, 3200 MT/s, 64 threads were used for the following evaluation. Time at 200 dimensionality (typical for the *E. coli* dataset) was 7.96 minutes to complete 100 FastICA runs, 1.10 minutes for distance matrix and clustering calculation, and 16.02 seconds for processing the final matrices. The resulting file sizes were 2.2 GB of temporary files, 13.3 MB for the final M matrix, and 2.9 MB for the final A matrix. Results for other dimensionalities are shown in **Note C in S1 Text**.

## Results

Here, we demonstrate how to build the iModulon structure of *Bacillus subtilis* from publicly available RNA-seq datasets using the workflow (**Fig 1A**) and characterize the iModulons with Pymodulon. All code to reproduce these results is available at: https://github.com/SBRG/iModulonMiner.

### Results from Steps 1 and 2: Compilation and processing of all publicly available RNA-seq datasets for *B. subtilis*

We compile the metadata for all publicly available RNA-seq data for *B. subtilis* in NCBI SRA (https://github.com/sbrg/iModulonMiner/tree/main/1_download_metadata). Here we utilize a dataset of 718 samples labeled as *Bacillus subtilis* RNA-seq data.

The *B. subtilis* dataset was subsequently processed using the RNA-seq pipeline available at https://github.com/sbrg/iModulonMiner/tree/main/2_process_data (**Fig D in S1 Text**). Ten samples failed to complete the processing pipeline, resulting in expression counts for 708 datasets.

### Results from Step 3: Quality control, metadata curation, and normalization

The *B. subtilis* compendium was subjected to five quality control criteria (**Fig 1B**). During manual curation, we removed some non-traditional RNA-seq datasets, such as TermSeq or RiboSeq. The final high-quality *B. subtilis* compendium contained 265 RNA-seq datasets (**Fig 1C**). Although manual curation is the most time-consuming part of the workflow, it facilitates deep characterization of patterns in the gene expression compendium. For example, application of Principal Component Analysis (PCA) to the *B. subtilis* expression compendium revealed that a large portion of the expression variation could be explained by the growth stage (**Fig 1D**). Finally, the log-TPM data within each project was centered to a project-specific reference condition (**Fig 1E**).

### Results from Step 4: Running independent component analysis

The *optICA* script (https://github.com/sbrg/iModulonMiner/tree/main/4_optICA) computes the optimal set of independent components and their activities (**Note D in S1 Text**). We apply a threshold to each independent component (Design and Implementation), resulting in gene sets called iModulons. This process resulted in 72 iModulons for the *B. subtilis* compendium that explained 67% of the expression variance in the compendium (**Fig E in S1 Text**).

### Results from Step 5: Characterizing iModulons

Here, we describe how the contents of the PyModulon package contributes to understanding information encoded in iModulons.

### iModulons are defined and grouped into categories based on annotation

The *IcaData* object, which houses all the relevant information about the identified iModulons, is generated (**Fig 2A**). Each independent component from ICA contains a gene weight for every gene in the genome. Only genes with weights above a specific threshold are considered to be in an iModulon. (**Fig 2B**). All thresholds are computed during initialization of the *IcaData* object (**Supplementary Methods in S1 Text**). Individual thresholds can be adjusted using the *change_threshold* function.

The *compute_trn_enrichments* function automatically identifies iModulons that significantly overlap with regulons found in the literature. The method can be used to search for simple regulons (i.e., groups of genes regulated by a single regulator) or complex regulons (i.e., groups of genes regulated by a combination of regulators). This method is built on top of the *compute_annotation_enrichment* method, which can be used for gene set enrichment analysis against any gene set, such as gene ontology terms, KEGG pathways, plasmids, or phages. Annotating iModulons typically results in their categorization into one of four classes: regulatory, functional, single-gene, or uncharacterized.

Of the 72 *B. subtilis* iModulons, 52 iModulons represented the effects of known transcriptional regulators. Together, these *Regulatory* iModulons explain 57% of the variance in the dataset (**Fig 2C**). The iModulon recall and regulon recall can be used to assess the accuracy of regulator enrichments (**Fig 2D**). The iModulon recall is the fraction of the iModulon that is part of the pre-defined regulon from the literature, whereas the regulon recall is the fraction of the regulon that is captured by the iModulon. iModulons in the top left-quadrant often represent subsets of known regulons. For example, there are three iModulons that each capture different subsets of the SigB regulon (**Fig 2E**). Even though only 28 of the 58 genes (48%) in the ResD iModulon have published ResD binding sites, we identified a conserved 16 base pair motif upstream of all 58 genes in the iModulon (**Fig 2F**).

Five additional iModulons were dominated by a single, high-coefficient gene, and are automatically identified by the method *find_single_gene_imodulons*. These *Single Gene (SG)* iModulons may arise from over-decomposition of the dataset [27,48] or artificial knock-out or overexpression of single genes. Together, these iModulons contribute to 1% of the variance.

The remaining 15 iModulons that could not be mapped to regulons present likely targets for the discovery of new regulons. Of those, the strongest candidates are the nine *Functional* iModulons, or iModulons that could be assigned a putative function. For example, one iModulon contains five genes in the same operon: *yvaC*, *yvaD*, *yvaE*, *yvaF*, and *azoRB*. Since YvaF is a putative transcription factor, we hypothesize that this iModulon is controlled by YvaF. Six *Uncharacterized* iModulons primarily contained either uncharacterized or unrelated genes, and contributed to 2% of the variance in the dataset.

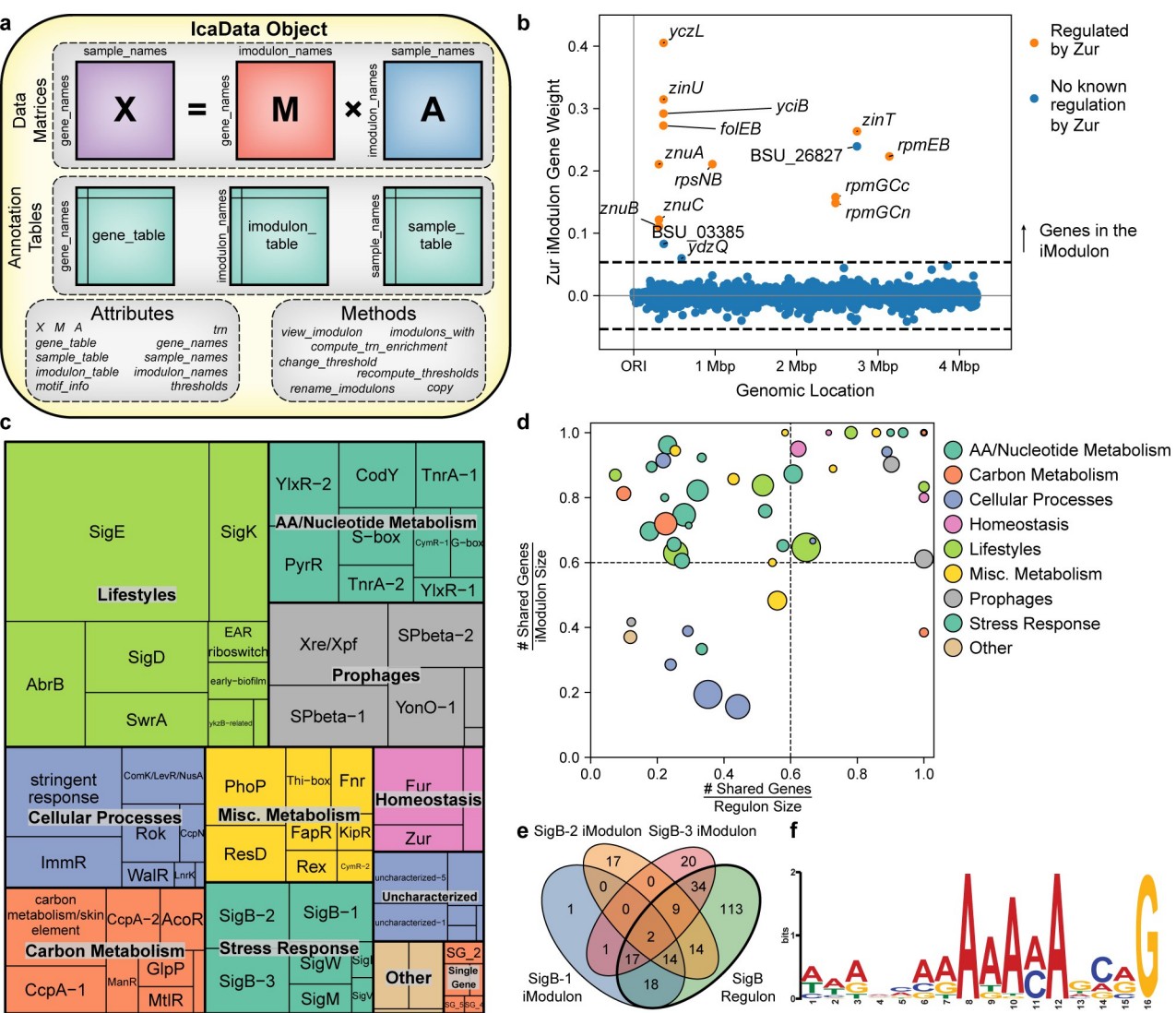

**Fig 2. Overview of the *B. subtilis* iModulon structure.** (a) Graphical representation of the *IcaData* object from PyModulon, illustrating the data, attributes, and methods stored in the object. (b) Example of an iModulon. Each point represents a gene. The x-axis shows the location of the gene in the genome, and the y-axis measures the weight of the gene in the Zur iModulon. Genes with prior evidence of Zur regulation are highlighted in orange. Genes outside the dashed black line are members of the Zur iModulon, whereas the genes inside the dashed black lines are not in the Zur iModulon. (c) Treemap of the 72 *B. subtilis* iModulons. The size of each box represents the fraction of expression variance that is explained by the iModulon. (d) Scatter plot comparing the overlap of each iModulon and its associated regulon(s). The circle size scales with the number of genes in the iModulon, and the color indicates the general category of the iModulon. (e) Venn diagram between the three SigB iModulons and the SigB regulon. (f) Motif identified upstream of all 58 genes in the ResD iModulon.

Altogether, these 72 iModulons provide a quantitative framework for understanding the TRN of *B. subtilis*. This framework can be used to both re-interpret previously published studies in the context of the full compendium, and to rapidly analyze new data.

## iModulon visualization empowers data exploration

PyModulon contains a suite of functions to create informative visualization, as described here: https://pymodulon.readthedocs.io/en/latest/tutorials/plotting_functions.html. One such function computes a clustered heatmap of iModulon activities to identify correlated groups of

iModulons (**Fig F in S1 Text**). These iModulons often respond to a common stimulus, and represent a computational method to define stimulons [52,53]. Here, we present two case studies that develop hypotheses of regulatory mechanisms based on iModulon visualizations.

First, we identified an uncharacterized iModulon that contains genes responsible for capsular polyglutamate synthesis, biofilm components, and synthesis of the peptide/polyketide antibiotic bacillaene [54] (**Fig 3A**). This iModulon is activated in early biofilm production and stationary phase (**Fig 3B**). As no single regulator is known to control all of these processes, this iModulon presents a hypothesis of the existence of a novel global regulator of biofilm formation.

Second, we examine three iModulons that contain three distinct sections of the *B. subtilis* prophage SPβ (**Fig 3C**), one of which coincided with nearly all genes known to be transcribed by YonO, a recently discovered single subunit phage RNA polymerase [55]. The SPbeta-1 and SPbeta-2 iModulons diverge in a single experiment, where *B. subtilis* was infected with either the phage Phi3T or SPβ [56] (**Fig 3D**). The activities of the YonO-1 iModulon are nearly identical to the SPbeta-2 iModulon (**Fig 3E**). However, the two major differences include YonO mutant strains [55] and a dataset where *B. subtilis* was exposed to heat shock at 53C [57]. The abnormally low YonO-1 iModulon activities are expected from the YonO mutant strain;

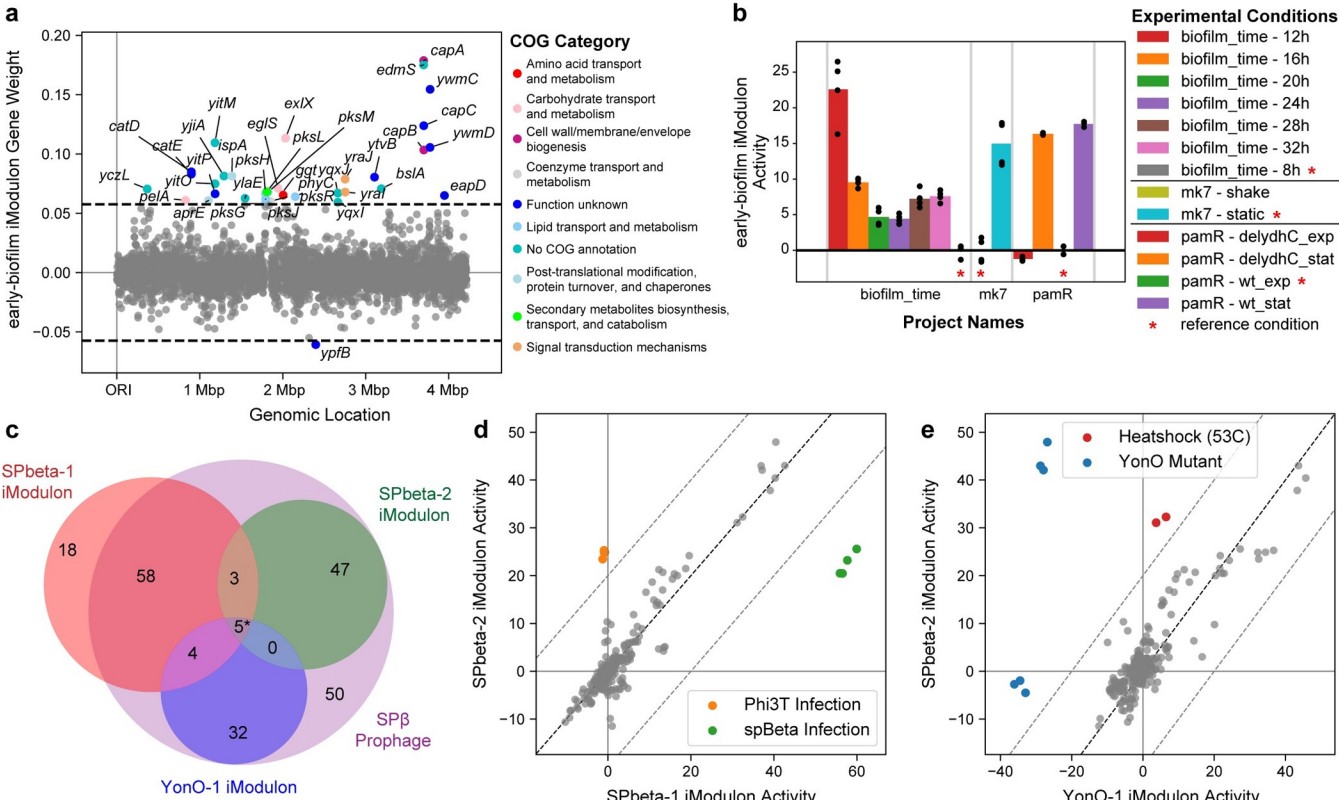

**Fig 3. Examples of insights derived from iModulons. (a)** Scatter plot of the gene weights in the newly discovered early-biofilm iModulon, created using the *plot_gene_weights* function. Genes outside the horizontal dashed black lines are in the iModulon, and genes are colored by their Cluster of Orthologous Gene (COG) category. **(b)** Bar plot of the iModulon activities for the early-biofilm iModulon, created from the *plot_activities* function. Individual points show iModulon activities for replicates, whereas the bars show the average activity for an experimental condition. Asterisks indicate the reference condition for each project. **(c)** Venn diagram comparing the SPbeta-1, SPbeta-2 and YonO-1 iModulons against the genes in the SPβ prophage. The asterisk indicates that one gene (*yozZ*) was in all three iModulons, but not in the prophage. **(d)** Scatter plot comparing the SPbeta-1 and SPbeta-2 iModulon activities, created from the *compare_activities* function. Each point represents a gene expression dataset under a specific condition. The center diagonal line is the 45-degree line of equal activities. **(e)** Scatter plot comparing the SPbeta-2 and YonO-1 iModulon activities. Each point represents a gene expression dataset under a specific condition.

however, the low activity during heat shock may indicate that the phage RNA polymerase YonO may be more sensitive to heat shock than the main *B. subtilis* RNA polymerase.

## Comparing iModulon structures across datasets and organisms reveals robustness

The differences in dataset and experimental conditions can create different iModulons within the same organism (see **Note E in S1 Text**), but previous studies have shown that similar iModulons can be found across disparate datasets [24,25]. To demonstrate this property, we use the *compare_ica* method to map the similarities between the iModulon structure presented here and an iModulon structure computed from a single microarray dataset [10,58]. The similarities are defined by the Pearson R correlation between the independent component gene weights, and are represented by the thickness of the arrows in **Fig G in S1 Text**. Of the 72 iModulons extracted from the RNA-seq compendium, 47 iModulons (65%) were highly similar to the microarray iModulons (**Fig 4A**). For example, nearly every gene in the Zur iModulon has nearly identical gene weights in both datasets (**Fig 4B**).

Presence of iModulons in two disparate datasets lends confidence of the biological significance of the component [24]. For example, an iModulon containing many uncharacterized genes was found in both datasets (**Fig 4C**). This is the same uncharacterized iModulon

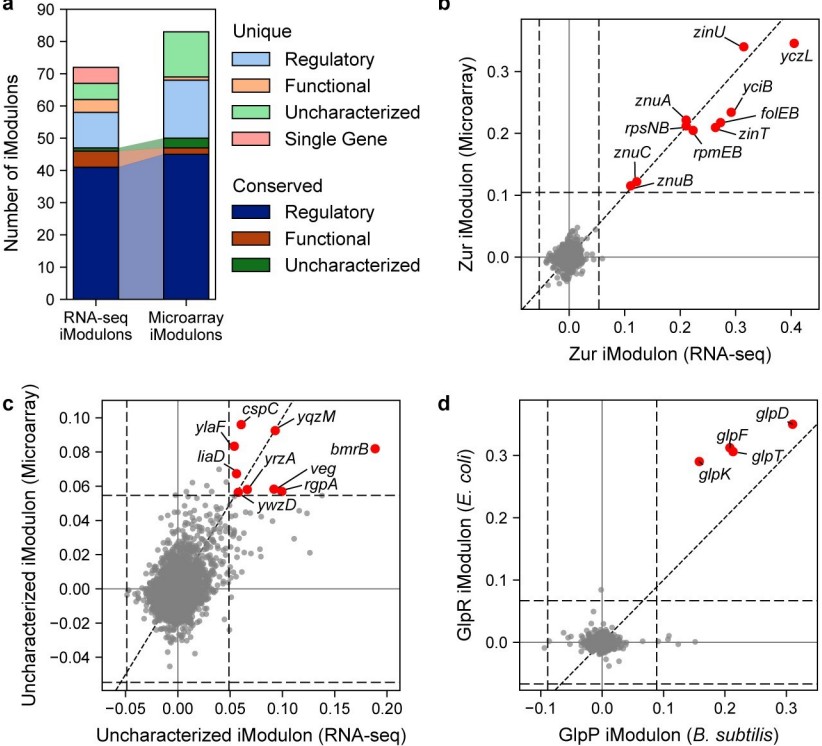

**Fig 4. Comparison of iModulon structures between datasets.** (a) Bar chart comparing the iModulons found in this RNA-seq dataset compared to a previous microarray dataset, colored by the type of iModulon. iModulons that are conserved between the two datasets are shown in a darker color. (b-d) Scatter plots comparing gene weights of iModulons found in different datasets, created using the *compare_gene_weights* function. Horizontal and vertical dashed lines indicate iModulon thresholds. Diagonal dashed line indicates the 45-degree line of equal gene weights. Genes in red are members of both iModulons. (b) Comparison of the Zur iModulon gene weights computed from the RNA-seq and microarray datasets. (c) Comparison of an uncharacterized iModulon found in both the RNA-seq and microarray datasets. (d) Comparison of the *B. subtilis* GlpP iModulon to the *E. coli* GlpR iModulon.

(uncharacterized-5) that was activated in the first few days of biofilm development (**Fig H in S1 Text**). In the microarray dataset, this uncharacterized iModulon was downregulated in late sporulation. This observation supports that the genes in this iModulon are likely co-regulated by a transcriptional regulator related to biofilm development.

In addition, iModulons can be compared between organisms using gene orthology. We compared the *B. subtilis* iModulon structure to a previously published *E. coli* iModulon structure [9], and found many orthologous iModulons (defined as iModulons containing orthologous genes with similar gene coefficients). We identified 22 iModulons in the *B. subtilis* dataset that were orthologous to *E. coli* iModulons (**Fig I in S1 Text**). For example, the weights of the genes in the *B. subtilis* GlpP iModulon were nearly identical to their orthologs in the *E. coli* GlpR iModulon, indicating that these genes are modulated in similar ratios across the two organisms (**Fig 4D**).

### The iModulonDB web page hosts iModulon analysis results

The results for the *Bacillus subtilis* dataset discussed here are available at https://iModulonDB. org/dataset.html?organism=b_subtilis&dataset=modulome [59].

### Availability and future directions

We have described two complementary tools to compile and explore iModulons. First, we present a GitHub repository that walks through each analysis in this manuscript (https:// github.com/sbrg/iModulonMiner). The pipeline is modular, as any step can be replaced with an alternative process, and the code in the repository can be modified for any new organism of interest. Second, we present PyModulon, a Python package for exploring iModulon properties, enrichments, and activities (https://pymodulon.readthedocs.io/en/latest/). We foresee that this workflow will be broadly applied to all publicly available datasets, resulting in a database of iModulons for every organism with sufficient data.

### Supporting information

**S1 Text.** Supplementary Information file that contains Supplementary Methods, Results, Notes A-E, Figs A-K and References.
(PDF)

### Acknowledgments

The authors would like to thank Dr. Amitesh Anand, Dr. Hyungyu Li, Dr. Henrique Machado, and Jayanth Krishnan for informative discussions. This research used resources of the National Energy Research Scientific Computing Center, a DOE Office of Science User Facility supported by the Office of Science of the U.S. Department of Energy under Contract No. DE-AC02-05CH11231.

### Author Contributions

**Conceptualization:** Anand V. Sastry, Saugat Poudel, Yara Seif.

**Funding acquisition:** Bernhard O. Palsson.

**Methodology:** Anand V. Sastry, Yuan Yuan, Saugat Poudel.

**Project administration:** Anand V. Sastry, Bernhard O. Palsson, Daniel C. Zielinski.

**Software:** Anand V. Sastry, Yuan Yuan, Saugat Poudel, Kevin Rychel, Reo Yoo, Cameron R. Lamoureux, Gaoyuan Li, Joshua T. Burrows, Siddharth Chauhan, Zachary B. Haiman, Tahani Al Bulushi.

**Supervision:** Anand V. Sastry, Bernhard O. Palsson, Daniel C. Zielinski.

**Writing – original draft:** Anand V. Sastry, Yuan Yuan, Kevin Rychel, Daniel C. Zielinski.

**Writing – review & editing:** Anand V. Sastry, Yuan Yuan, Saugat Poudel, Kevin Rychel, Reo Yoo, Cameron R. Lamoureux, Gaoyuan Li, Joshua T. Burrows, Siddharth Chauhan, Zachary B. Haiman, Tahani Al Bulushi, Yara Seif, Bernhard O. Palsson, Daniel C. Zielinski.

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
