## [Decision Letter · Decision Letter 0]

21 Jul 2024

Dear Dr. Zielinski,

Thank you very much for submitting your manuscript "iModulonMiner and PyModulon: Software for unsupervised mining of gene expression compendia" for consideration at PLOS Computational Biology.

As with all papers reviewed by the journal, your manuscript was reviewed by members of the editorial board and by several independent reviewers. In light of the reviews (below this email), we would like to invite the resubmission of a significantly-revised version that takes into account the reviewers' comments.

Overall the reviews were very positive on the utility of the tool and the clarity of the manuscript and documentation. However, reviews indicated several changes and clarifications that would improve the usability of the tool, particularly for general users.

We cannot make any decision about publication until we have seen the revised manuscript and your response to the reviewers' comments. Your revised manuscript is also likely to be sent to reviewers for further evaluation.

Sincerely,

Nic Vega, Ph.D.

Academic Editor

PLOS Computational Biology

Stacey Finley, Ph.D.

Section Editor

PLOS Computational Biology

Overall the reviews were very positive on the utility of the tool and the clarity of the manuscript and documentation. However, reviews indicated several changes and clarifications that would improve the usability of the tool, particularly for general users.

Reviewer's Responses to Questions

**Comments to the Authors:**

Reviewer #1: the review is uploaded as an attachment.

Reviewer #2: The iModulonMiner pipeline processes RNA-seq data through Independent Component Analysis to identify co-regulated gene sets. Applied to Bacillus subtilis, it predicts regulatory interactions, identifies gene groups regulated by unknown factors, and characterizes a novel phage RNA polymerase. The accompanying PyModulon Python package aids visualization and exploration.

Strengths:

+ Well documented source code and python packages

+ Results demonstrated on real organism

+ Automated scripts

Recommendations:

1) What is the effect of number of varying conditions of organisms on the overall pipeline? Is it possible to demonstrate that with an example and comparison to a baseline?

2) In few of the steps ablations might be possible or be included as option in the pipeline and overall effect may be different. For example, alternatives to ICA (3rd step) and nf-core or STAR (second step) are not explored. Why?

Reviewer #3: The manuscript presented a five-step computational pipeline, called iModulonMiner, to generate the iModulon structure of any organism based on RNA-seq data and a Python package, PyModulon, to characterize, visualize, and explore computed iModulons. They used the pipeline to calculate imodulons for Bacillus subtilis and compared the results with that of a previous study using microarray data to show the robustness of the method. The pipeline and package are useful for scientists who want to discover novel regulatory interactions from large scale transcriptomics data. However, some details are still missing in the manuscript and the github files, making it difficult to repeat the whole calculation procee.

1. The authors stated that in the third step, the data must be inspected to ensure quality, and curated to include all appropriate metadata. This is actually a time-consuming manual work and rely heavily on the expertise of the curator. The authors may provide more detail instructions on how to do it properly to ensure the reliability of the calculated modulons. They may add the instructions in the B. subtilis results section so that readers can actually follow the steps to perform the analysis using the B. subtilis data.

2. More data should be added for the pipeline such as the calculation time in a typical computer and the data and file sizes at each step.

3. I have tried to use the PyModulon to repeat the calculations but failed at step 2. I always got the “nextflow: command not found” error message when I run “nextflow run main.nf -profile local --organism bacillus_subtilis --metadata ../test/test_metadata.tsv --sequence_dir ../test/sequence_files/ --outdir ../test/nf_results/” command. Even though they said “install the dependencies in requirements.txt” in the github readme file, I actually could not find the requirements.txt file in the iModulonMiner project. The file required for generating Gene table and KEGG/GO annotations, “gene_annotation.ipynb”, is also missing.

4. More discussion on the comparison of the 72 imodulons obtained from the new pipeline and the 83 imodulons from their previous study (ref 10) should be added. The manuscript mentioned that 47 iModulons (65%) were highly similar but how the similarity was quantitatively defined was not provided. I am not sure if this number is enough to prove the calculation results is robust. There is no discussion on the 25 imodulons (36 from previous study) which are not similar. The author may need to provide the complete data in a supplementary file for further biological examination.

**Have the authors made all data and (if applicable) computational code underlying the findings in their manuscript fully available?**

Reviewer #1: Yes

Reviewer #2: Yes

Reviewer #3: Yes

PLOS authors have the option to publish the peer review history of their article (what does this mean?). If published, this will include your full peer review and any attached files.

Reviewer #1: No

Reviewer #2: No

Reviewer #3: No
---

## [Decision Letter · Decision Letter 1]

9 Oct 2024

Dear Dr. Zielinski,

We are pleased to inform you that your manuscript 'iModulonMiner and PyModulon: Software for unsupervised mining of gene expression compendia' has been provisionally accepted for publication in PLOS Computational Biology.

Best regards,

Nic Vega, Ph.D.

Academic Editor

PLOS Computational Biology

Stacey Finley, Ph.D.

Section Editor

PLOS Computational Biology

Reviewer's Responses to Questions

**Comments to the Authors:**

Reviewer #1: The original submission was impressive, and the revisions have elevated the article even further. I strongly recommend accepting this manuscript for publication in PLOS Computational Biology. It's a significant contribution with a well-constructed argument, clear writing, and excellent visuals. The tables, supplementary materials, and responses to reviewer comments demonstrate thoroughness and consideration. I appreciate the authors' thoughtful approach to the revision process. I'm eager to see this paper in its published form.

Reviewer #2: The revision addressed the concerns.

Reviewer #3: All the comments have been properly addressed and I do not have any further comments.

**Have the authors made all data and (if applicable) computational code underlying the findings in their manuscript fully available?**

Reviewer #1: Yes

Reviewer #2: None

Reviewer #3: Yes

PLOS authors have the option to publish the peer review history of their article (what does this mean?). If published, this will include your full peer review and any attached files.

Reviewer #1: No

Reviewer #2: No

Reviewer #3: **Yes: **Hongwu Ma

---

## [Editor Report · Acceptance letter]

17 Oct 2024

PCOMPBIOL-D-24-00592R1 

iModulonMiner and PyModulon: Software for unsupervised mining of gene expression compendia

Dear Dr Zielinski,

I am pleased to inform you that your manuscript has been formally accepted for publication in PLOS Computational Biology. Your manuscript is now with our production department and you will be notified of the publication date in due course.

With kind regards,

Dorothy Lannert
